



# Graphics processing unit accelerated ice flow solver for unstructured meshes using the Shallow Shelf Approximation (FastIceFlo v1.0)

Anjali Sandip[1], Ludovic Räss[2,3], and Mathieu Morlighem[4]

[1]Department of Mechanical Engineering, University of North Dakota, North Dakota, USA
[2]Laboratory of Hydraulics, Hydrology and Glaciology (VAW), ETH Zurich, Zurich, Switzerland
[3]Swiss Federal Institute for Forest, Snow and Landscape Research (WSL), Birmensdorf, Switzerland
[4]Department of Earth Sciences, Dartmouth College, New Hampshire, USA

**Correspondence:** Anjali Sandip (anjali.sandip@und.edu)

**Abstract.** Ice-sheet flow models capable of accurately projecting their future mass balance constitute tools to improve flood risk assessment and assist sea-level rise mitigation associated with enhanced ice discharge. Some processes that need to be captured, such as grounding line migration, require high spatial resolution (1 km or better). Conventional ice flow models may need significant computational resources because these models mainly execute Central Processing Units (CPUs), which lack massive parallelism capabilities and feature limited peak memory bandwidth. On the other side of the spectrum, Graphics Processing Units (GPUs) are ideally suited for high spatial resolution as the calculations at every grid point can be performed concurrently by thousands of threads or parallel workers. In this study, we combine GPUs with the pseudo-transient (PT) method, an accelerated iterative and matrix-free solving approach, and investigate its performance for finite elements and unstructured meshes applied to two-dimensional (2-D) models of real glaciers at a regional scale. For both Jakobshavn and Pine Island glacier models, the number of nonlinear PT iterations to converge for a given number of vertices $N$ scales in the order of $\mathcal{O}(N^{1.2})$ or better. We compared the performance of PT CUDA C implementation with a standard finite-element CPU-based implementation using the metric: price and power consumption to performance. The single Tesla V100 GPU is 1.5 times the price of the two Intel Xeon Gold 6140 CPU processors. The power consumption of the PT CUDA C implementation was approximately one-seventh of the standard CPU implementation for the test cases chosen in this study. We expect a minimum speed-up of >1.5 to justify the Tesla V100 GPU price to performance. We report the performance (or the speed-up) across glacier configurations for degrees of freedom (DoFs) tested to be >1.5 on a Tesla V100. This study is a first step toward leveraging GPU processing power for accurate polar ice discharge predictions. The insights gained from this study will benefit efforts to diminish spatial resolution constraints at increased computing speed. The increased computing speed will allow running ensembles of ice-sheet flow simulations at the continental scale and at high resolution, previously not possible, enabling quantification of model sensitivity to changes in future climate forcings. These findings will be significantly beneficial for process-oriented and sea-level-projection studies over the coming decades.





## 1 Introduction

Global mean sea level is rising at an average rate of 3.7 mm yr$^{-1}$, posing a significant threat to coastal communities and global ecosystems (Hinkel et al., 2014; Kopp et al., 2016). The increase in ice discharge from the Greenland and Antarctic ice sheets

significantly contributes to sea level rise. However, their dynamic response to climate change remains a fundamental uncertainty in future projection (Rietbroek et al., 2016; Chen et al., 2017; IPCC, 2021). While much progress has been made over the last decades, several critical physical processes remain poorly known (Pattyn and Morlighem, 2020). Existing computational resources limit the spatial resolution and simulation time on which continental-scale ice-sheet models can run. Some processes, such as grounding line or ice front dynamics, require spatial resolutions of 1 km or better (Larour et al., 2012; Aschwanden

et al., 2021; Castleman et al., 2022). Most numerical models are designed for central processing units (CPU). The performance of CPUs is slowly leveling off. It remains to be seen whether high-resolution modeling will be available at the continental scale (or ice-sheet scale), especially for complex flow models, such as Full-Stokes, which remain challenging to employ beyond the regional scale. On the other side of the spectrum, graphics processing units (GPUs) have been booming over the past decade (Brædstrup et al., 2014; Häfner et al., 2021). Developing algorithms and solvers to leverage GPU computing power has become

an essential aspect of numerical computing.

However, the traditional way of solving the governing equations of ice-sheet flow, such as the finite-element analysis, is not adapted to GPUs as they cannot handle large sparse matrices and linear solvers. Räss et al. (2020) proposed an alternative approach by re-formulating the flow equations in the form of a pseudo-transient (PT) model. The PT method augments physically motivated time-dependent inertial terms to targeted time-independent equations, which is the case of the ice sheet

stress balance. These equations are explicitly and iteratively updated in pseudo-time $\tau$ until we reach a steady state, which provides a solution to the initial time-independent equations. The explicit update eliminates the need for an expensive stiffness matrix assembly, making it matrix-free and suitable for parallel implementations and most operations would be identical for each grid point (Frankel, 1950; Poliakov et al., 1993; Kelley and Liao, 2013). Räss et al. (2020) introduced this method and combined its implementation with GPUs to enable the development of high spatial resolution full Stokes ice-sheet flow models,

two-dimensional (2-D) and three-dimensional (3-D), on uniform grids (Räss et al., 2020). Among the study's limitations are the proposed finite differences numerical scheme, the uniform structured grid, and the applications limited to simple idealized cases.

Here, we extend the accelerated PT method to finite elements on unstructured meshes. We developed a CUDA C implementation of the PT depth-integrated Shallow Shelf Approximation (SSA) and apply it to real glaciers at a regional scale: Pine Island

Glacier, in west Antarctica, and Jakobshavn Isbræ, in western Greenland. We compare the PT CUDA C implementation with the more standard finite-element CPU implementation available in the Ice-sheet and Sea-level System Model (ISSM) using the same mesh, model equations, and boundary conditions. In Section 2, we present the mathematical reformulation of the 2D SSA momentum balance equations to incorporate the PT method followed by its weak formulation and spatial discretization. Section 3 describes the numerical experiments conducted, chosen glacier model configurations, hardware implementation, and





performance assessment metrics. In Sections 4 and 5, we illustrate the method's performance and conclude with future research
      directions.

## 2    Methods

### 2.1    Mathematical formulation of 2D SSA model

We employ SSA (MacAyeal, 1989)) to solve the momentum balance. The SSA equations in the matrix form read :

$$\nabla \cdot (2H\mu\dot{\boldsymbol{\varepsilon}}_{\text{SSA}}) = \rho g H \nabla \mathbf{s} + \alpha^{\mathbf{2}}\mathbf{v} \tag{1}$$

where $\dot{\boldsymbol{\varepsilon}}_{\text{SSA}}$ is defined as

$$\dot{\boldsymbol{\varepsilon}}_{\text{SSA}} = \begin{pmatrix} 2\dot{\varepsilon}_{xx} + \dot{\varepsilon}_{yy} & \dot{\varepsilon}_{xy} \\ \\ \dot{\varepsilon}_{xy} & 2\dot{\varepsilon}_{yy} + \dot{\varepsilon}_{xx} \end{pmatrix} \tag{2}$$

$v_x$ and $v_y$ are the $x$ and $y$ ice velocity components, $H$ is the ice thickness, $\rho$ the ice density, $g$ the gravitational acceleration, $s$ glacier's upper surface z-coordinate and $\alpha^2\mathbf{v}$ is the basal friction term. The ice viscosity $\mu$ follows Glen's flow law (Glen, 1955):

$$\mu = \frac{B}{2\,\dot{\varepsilon}_e^{(n-1)/n}} \tag{3}$$

where B is ice rigidity, $\dot{\varepsilon}_e$ is the effective strain-rate, and $n = 3$ is Glen's power-law exponent.

In terms of boundary conditions, we apply water pressure at the ice front $\Gamma_\sigma$, and non-homogeneous Dirichlet boundary conditions on the other boundaries $\Gamma_u$ (based on observed velocity).

### 2.2    Mathematical reformulation of 2D SSA model to incorporate PT method

The solution to Eq. (1) is commonly achieved by using the finite-element or finite-difference method. These methods require to spatially discretize the domain and build a linear system of equations that is then solved using either a direct or iterative solver. Robust matrix-based solvers exhibit significant scaling limitations restricting their applicability when considering high-resolution or three-dimensional configurations. Iterative solving approaches permit circumventing most of the scaling limitations. The challenge is instead to prevent the iteration count from growing exponentially. We propose the accelerated pseudo-transient (PT) method as an alternative approach. The method augments the steady-state viscous flow equations (1) by adding a transient term. One can then use the transient term to integrate the equations in pseudo-time $\tau$, seeking an implicit solution once the steady state is reached at $\tau \to \infty$.

Building upon work from previous studies (Omlin et al., 2018; Duretz et al., 2019; Räss et al., 2019), we reformulate the 2-D SSA steady-state momentum balance equations to incorporate the usually ignored inertial terms:

$$\nabla \cdot (2H\mu\dot{\boldsymbol{\varepsilon}}_{\text{SSA}}) - \rho g H \nabla s - \alpha^2\mathbf{v} = \rho H \frac{\partial \mathbf{v}}{\partial \tau} \tag{4}$$





allows us to turn the steady-state equations into transient diffusion of velocities $v_{x,y}$. The velocity time derivatives represent physically motivated expressions we can further use to iteratively reach a steady-state, which provides the solution of the original time-independent equations. Since we are here only interested in the steady-state, transient processes evolve in numerical

pseudo-time $\tau$:

$$
\begin{aligned}
\rho H \frac{\partial v_x}{\partial \tau} &= R_x \\
\rho H \frac{\partial v_y}{\partial \tau} &= R_y
\end{aligned}
\tag{5}
$$

where $R_x$ and $R_y$ correspond to the right-hand-side expressions of Eq. (1) and define the residuals of the original SSA equations we seek a solution for. We define the transient *pseudo* time step $\Delta \tau$ as a field variable, spatially variable, chosen to minimize

the number of nonlinear PT iterations.

### 2.3 Pseudo-time stepping

We can advance in numerical pseudo-time using a forward Euler method. We choose our inertial term by approximating the transient diffusive system for both $v_x$ and $v_y$,

$$
\begin{aligned}
\rho H \frac{\partial v_x}{\partial \tau} &= \frac{\partial}{\partial x}\left(4 H \mu \frac{\partial v_x}{\partial x}\right) \\
\rho H \frac{\partial v_y}{\partial \tau} &= \frac{\partial}{\partial y}\left(4 H \mu \frac{\partial v_y}{\partial y}\right)
\end{aligned}
\tag{6}
$$

where one recognises the diffusion of $v_{x,y}$ and the effective dynamic viscosity $4\mu/\rho$ as diffusion coefficient. Using the diffusion analogy, we can use this information to define the CFL-like (Courant-Friedrich-Lewy) stability criterion for the PT iterative scheme. The explicit CFL time step for viscous flow is given by:

$$
\Delta \tau_{\max} = \rho \frac{\Delta x^2}{4\mu(1+\mu_b) \times n_{\dim}}
\tag{7}
$$

where $\Delta x$ is the grid spacing, $\mu_b$ is the numerical bulk ice viscosity and $n_{\dim} = 2.1, 4.1, 6.1$ in 1, 2 and 3D, respectively.

### 2.4 Viscosity continuation

We implement a continuation on the nonlinear strain-rate dependent effective viscosity $\mu_{\mathrm{eff}}$ to avoid the iterative solution process to diverge as strain-rate values may not satisfy the momentum balance at the beginning of the iterative process, and may thus be far from equilibrium. At every pseudo-time step, the effective viscosity $\mu_{\mathrm{eff}}$ is updated in the logarithmic space:

$$
\mu_{\mathrm{eff}} = \exp\left(\theta_\mu \log(\mu) + (1-\theta_\mu)\log(\mu_{\mathrm{eff}}^{\mathrm{old}})\right) ,
\tag{8}
$$

where the scalar $10^{-2} < \theta_\mu < 1$ is selected such that we provide sufficient time to relax the nonlinear viscosity at the start of the pseudo-iterative loop.



### 2.5 Acceleration owing to damping

The major limitation of this simple first-order, or Picard-type, iterative approach resides in the poor iteration count scaling with increased numerical resolution. The number of iterations needed to converge for a given problem for $N$ number of grid points involved in the computation, scales in the order of $\mathcal{O}(N^2)$.

To address this limitation, we consider a second-order method, referred to as the second-order Richardson method, as introduced by Frankel (1950). This approach allows to aggressively reduce the number of iterations to the number of grid points, making the method scale to $\approx \mathcal{O}(N^{1.2})$. Optimal scaling can be achieved by realizing that the PT framework's diffusion-type of updates readily provided can be divided into two wave-like update steps. Transitioning from diffusion to wave like pseudo-physics exhibits two main advantages: (i) the wave-like time step limiter is function of $\Delta x$ instead of $\Delta x^2$, and (ii) it is possible to turn the wave equation into a damped wave equation. The latter permits finding optimal tuning parameter to achieve optimal damping, resulting in fast convergence. Let's assume the following diffusion-like update step, reported here for the $x$ direction only:

$$\rho H \frac{\partial v_x}{\partial \tau} = \frac{\partial}{\partial x} 4H\mu \frac{\partial v_x}{\partial x} \ . \tag{9}$$

The above expression would result in the following update rule:

$$v_x = v_x^{\text{old}} + \frac{\Delta \tau_{\text{D}}}{\rho} \left( \frac{\partial}{\partial x} 4\mu \frac{\partial v_x}{\partial x} \right) \tag{10}$$

where $\Delta \tau_{\text{D}} \approx \Delta x^2/(4\mu/\rho)/4.1$ is the diffusion-like time step limiter. This system can be rewritten in wave-equation style:

$$A_x = A_x^{\text{old}} + \frac{\Delta \tau_{\text{w}}}{\rho} \left( \frac{\partial}{\partial x} 4\mu \frac{\partial v_x}{\partial x} \right)$$

$$v_x = v_x^{\text{old}} + \Delta \tau_{\text{w}} A_x \tag{11}$$

where $\Delta \tau_{\text{w}} \approx \Delta x / \sqrt{4\mu/\rho}/2.1$ is the wave-like time step limiter where one recognises a term analogous to the numerical wave velocity. Moreover, the Eq. (11) can be damped by adding friction $\gamma$ to the current $A_x$:

$$A_x = A_x^{\text{old}}(1-\gamma) + \frac{\Delta \tau_{\text{w}}}{\rho} \left( \frac{\partial}{\partial x} 4\mu \frac{\partial v_x}{\partial x} \right)$$

To maintain solution stability we include relaxation $\theta_v$:

$$v_x = v_x^{\text{old}} + \theta_v \Delta \tau_{\text{w}} A_x \tag{12}$$

where $0 < \gamma < 1$ and $0 < \theta_v < 1$.

Alternative and complementary details about the PT acceleration can be found in Räss et al. (2019), Duretz et al. (2019), Räss et al. (2020) while an in-depth analysis is provided in Räss et al. (2022).

### 2.6 Weak formulation and finite-element discretization

Using the PT method, the equations to solve are now:

$$\rho H \frac{\partial \mathbf{v}}{\partial \tau} = \nabla \cdot 2H\mu \dot{\boldsymbol{\varepsilon}}_{\text{SSA}} - \rho g H \nabla s - \alpha^2 \mathbf{v} \tag{13}$$





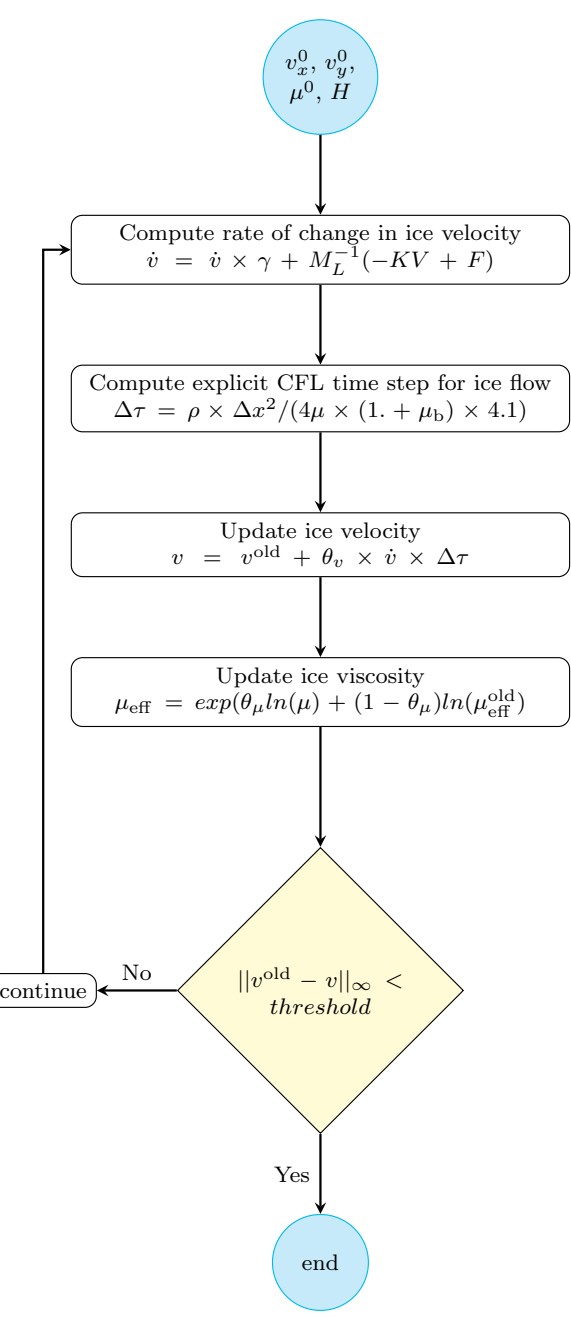

**Figure 1.** PT iterative algorithm for unstructured meshes applied to solve 2-D SSA momentum balance equations.

The weak form (assuming homogeneous Dirichlet conditions along all model boundaries for simplicity) is: $\forall \mathbf{w} \in \mathcal{H}^1(\Omega)$,

$$\int_\Omega \rho H \frac{\partial \mathbf{v}}{\partial \tau} \cdot \mathbf{w} \, d\Omega + \int_\Omega 2 H \mu \dot{\boldsymbol{\varepsilon}}_{\mathrm{SSA}} : \dot{\boldsymbol{\varepsilon}}_w \, d\Omega$$


$$= \int_\Omega -\rho g H \nabla s \cdot \mathbf{w} - \alpha^2 \mathbf{v} \cdot \mathbf{w} \, d\Omega \quad (14)$$



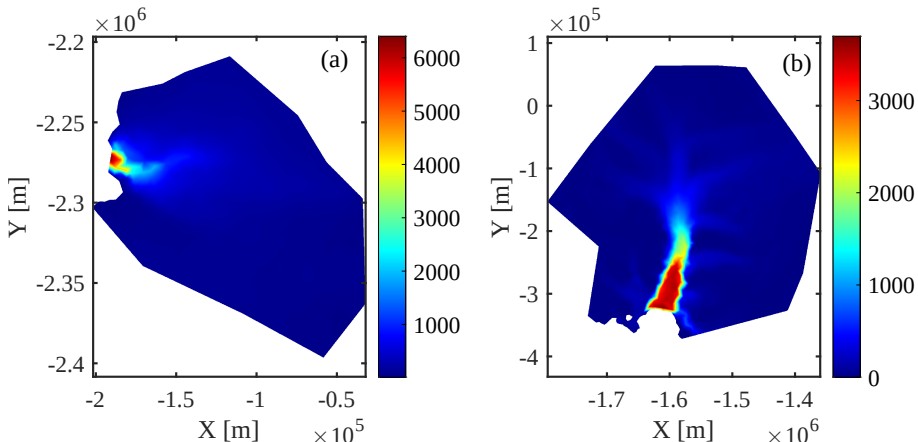

**Figure 2.** Glacier model configurations; observed surface velocities in m yr$^{-1}$ interpolated on a uniform mesh. Panels **(a)** and **(b)** correspond to Jakobshavn Isbræ and Pine Island Glacier respectively.

where $\mathcal{H}^1(\Omega)$ is the space of square-integrable functions whose first derivatives are also square integrable.

Once discretized using the finite-element method, the matrix system to solve is:

$$M\dot{\mathbf{V}} + K\mathbf{V} = \boldsymbol{F} \tag{15}$$

where $M$ is the mass matrix, $K$ is the stiffness matrix, $\boldsymbol{F}$ is the right hand side or load vector, and $\mathbf{V}$ is the vector of ice
velocity.

We can compute $\dot{\mathbf{V}}$ by solving:

$$\dot{\mathbf{V}} \simeq \mathbf{M}_L^{-1}(-K\mathbf{V} + \mathbf{F}) \tag{16}$$

where $\mathbf{M}_L$ a lumped mass matrix to avoid the resolution of a matrix system.

We hence have an explicit expression of the time derivative of the ice velocity for each vertex of the mesh:

$$\dot{v}_{xi} = \frac{1}{\rho H m_{Li}}\left(-\int_\Omega \left(4H\mu\frac{\partial v_x}{\partial x} + 2H\mu\frac{\partial v_y}{\partial y}\right)\frac{\partial \varphi_i}{\partial x}\right.$$

$$+\left(H\mu\frac{\partial v_x}{\partial y} + H\mu\frac{\partial v_y}{\partial x}\right)\frac{\partial \varphi_i}{\partial y}d\Omega$$

$$\left.+\int_\Omega -\rho g H\frac{\partial s}{\partial x}\varphi_i - \alpha^2 v_x\varphi_i\, d\Omega\right) \tag{17}$$





$$\dot{v}_{yi} = \frac{1}{\rho H m_{Li}} \left( -\int_\Omega \left( 4H\mu \frac{\partial v_y}{\partial y} + 2H\mu \frac{\partial v_x}{\partial x} \right) \frac{\partial \varphi_i}{\partial y} \right.$$

$$+ \left( H\mu \frac{\partial v_x}{\partial y} + H\mu \frac{\partial v_y}{\partial x} \right) \frac{\partial \varphi_i}{\partial x} d\Omega$$

$$\left. + \int_\Omega -\rho g H \frac{\partial s}{\partial y} \varphi_i - \alpha^2 v_y \varphi_i \, d\Omega \right) \quad (18)$$

where $m_{Li}$ is the component number $i$ along the diagonal of the lumped mass matrix $M_L$.

For every nonlinear PT iteration, we compute the rate of change in velocity $\dot{\mathbf{v}}$ and the explicit CFL time step $\Delta\tau$. We then
deploy the reformulated 2D SSA momentum balance equations to update ice velocity $\mathbf{v}$ followed by ice viscosity $\mu_{eff}$. We iterate in pseudo-time until the stopping criterion is met (Fig. 1).

## 3 Numerical experiments

### 3.1 Glacier model configurations

To test the performance of the PT method beyond simple idealized geometries, we apply it to two regional-scale glaciers:
Jakobshavn Isbræ, in western Greenland, and Pine Island Glacier, in west Antarctica (Fig. 2). For Jakobshavn Isbræ, we rely on BedMachine Greenland v4 (Morlighem et al., 2017) and also invert for basal friction to infer the basal boundary conditions. Note that the inversion is run on Ice-sheet and Sea-level System Model (ISSM), using a standard approach (Larour et al., 2012). For Pine Island Glacier, we initialize the ice geometry using BedMachine Antarctica v2 (Morlighem et al., 2020) and infer the friction coefficient using surface velocities derived from satellite interferometry (Rignot et al., 2011).

### 3.2 Hardware implementation

We developed a CUDA C implementation to solve the SSA equations using the PT approach on unstructured meshes. We choose a stopping criterion of $||v^{old} - v||_\infty < 10 \ m \ yr^{-1}$. The software solves the 2-D SSA momentum balance equations on a single GPU. We use here an NVIDIA Tesla V100 SXM2 GPU featuring 16 gigabytes (GB) onboard memory and an Ampere A100 SXM4 featuring 80GB onboard memory. We compare the PT Tesla V100 GPU implementation with ISSM's "standard"
CPU implementation using a conjugate gradient (CG) iterative solver. As a CPU, we used a 64-bit 18-core Intel Xeon Gold 6140 processor with 192 GB of RAM per vertex. We executed multi-core MPI-parallelized ice-sheet flow simulations on two CPUs, all 36 cores enabled(Larour et al., 2012; Habbal et al., 2017). We performed computations using double-precision arithmetic.



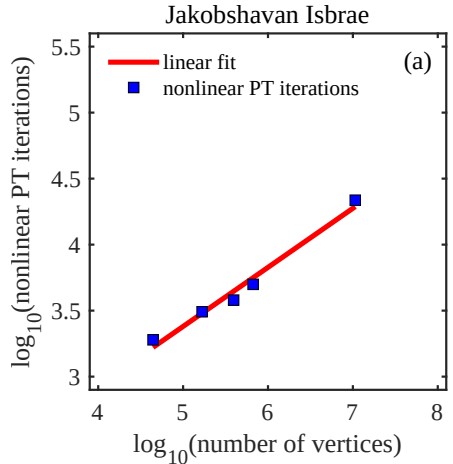
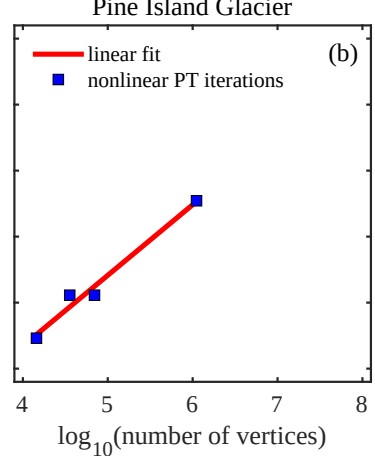

**Figure 3.** Performance assessment of PT CUDA C implementation for unstructured meshes.

### 3.3 Performance assessment metrics

To investigate the PT CUDA C implementation for unstructured meshes, we report the number of vertices (or grid size) and the corresponding number of nonlinear PT iterations needed to meet the stopping criterion.

We employ the computational time required to reach convergence as a proxy to assess and compare the performance of PT CUDA C with the ISSM CG CPU implementation. We ensure to exclude from timing all pre- and post-processing steps. We quantify the relative performance of the CPU and GPU implementations as the speed-up $S$, given by:

$$S = \frac{t_{\mathrm{CPU}}}{t_{\mathrm{GPU}}} \; . \tag{19}$$

The PT method applied to solve nonlinear momentum balance equations is a memory-bound algorithm (Räss et al., 2020); therefore, the wall time depends on the memory throughput. Hence, to assess the performance of PT CUDA C implementation developed in this study, in addition to speed-up, we employ the effective memory throughput metric (Räss et al., 2020, 2022), defined as:


$$T_{\mathrm{eff}} = \frac{n_{\mathrm{n}} \, n_{\mathrm{iter}} \, n_{\mathrm{IO}} \, n_{\mathrm{p}}}{1024^3 \, t_{\mathrm{n_{iter}}}} \; , \tag{20}$$

where $n_{\mathrm{n}}$ represents the total number of vertices, $n_{\mathrm{iter}}$ a given number of PT iterations, $n_{\mathrm{p}}$ the arithmetic precision, $t_{\mathrm{n_{iter}}}$ the time taken to complete $n_{\mathrm{iter}}$ iterations and $n_{\mathrm{IO}}$ the minimal number of non-redundant memory accesses (read/write operations). The number of reads and write operations needed for this study would be 8; update $v_x$, $v_y$, and nonlinear viscosity arrays for

every PT iteration, in addition to reading the basal friction coefficient and the masks.



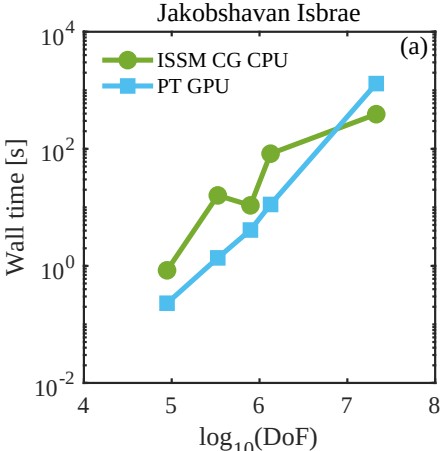
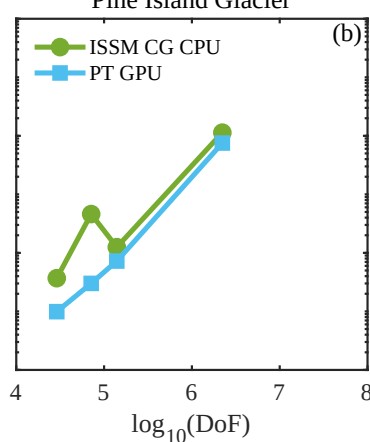

**Figure 4.** Performance comparison of PT Tesla V100 implementation with the CPU implementation employing wall time (or computational time to reach convergence). Note that wall time does not include pre- and post-processing steps.

.

## 4   Results and discussion

To investigate the PT CUDA C implementation for unstructured meshes, we report the number of vertices (or grid size) and the corresponding number of nonlinear PT iterations needed to meet the stopping criterion (Fig. 3). For both Jakobshavn and Pine Island glacier models, the number of nonlinear PT iterations to converge for a given number of vertices $N$ scales in the order of $\approx \mathcal{O}(N^{1.2})$ or better. We chose damping parameter $\gamma$, nonlinear viscosity relaxation scalar $\theta_\mu$, and transient *pseudo* time step $\Delta\tau$ to maintain the linear scaling described above. We observed an exception at $\sim 3e^7$ degrees of freedom (DoFs) for the Pine Island glacier model; optimal solver parameters are unidentifiable. We will investigate further the convergence for the Pine Island glacier model at $\sim 3e^7$ DoFs in the following steps. Among the two glacier models chosen in this study, for a given number of vertices $N$, Jakobshavn Isbræ resulted in faster convergence rates which we attribute to differences in scale and bed topography and the nonlinearity of the problem.

To assess and compare the PT GPU implementation with the traditional matrix-based CPU implementation, we employed price and power consumption to performance as a metric. The single Tesla V100 GPU is 1.5 times the price of the two Intel Xeon Gold 6140 CPU processors chosen in this study. The power consumption of the PT GPU implementation was measured using NVIDIA System Management Interface. For the range of DoFs tested, the power usage for both glacier configurations to meet the stopping criterion was $38 \pm 1$ W. Power consumption measurement for the CPU implementation was taken from the hardware specification sheet; thermal design power. For a 64-bit 18-core Intel Xeon Gold 6140 processor, the thermal design power is 140 W. We executed the CPU-based multi-core MPI-parallelized ice-sheet flow simulations on two CPUs, all 36 cores enabled, and we chose the power consumption to be 280 W. This is a first-order estimate. Thus the power consumption of the

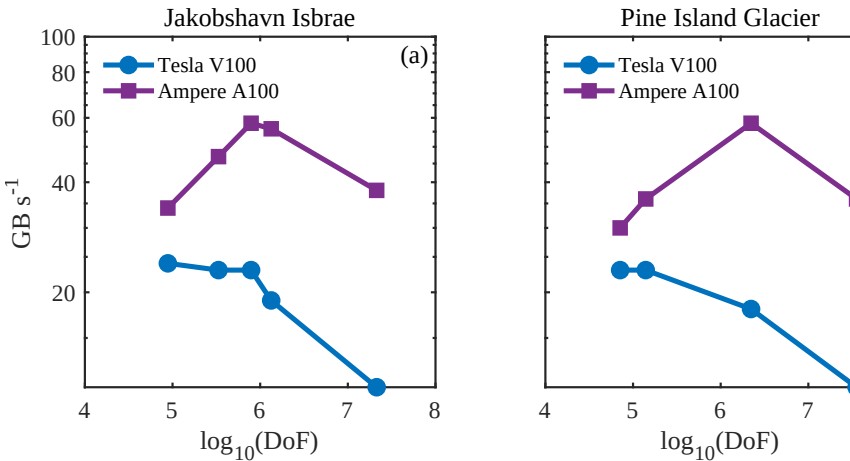

**Figure 5.** Performance assessment of PT CUDA C implementation across GPU architectures employing effective memory throughput.

.

PT GPU implementation was approximately one-seventh of the traditional CPU implementation for the test cases chosen in this study.

As described in Section 3.3, we quantify the relative performance of the CPU and GPU implementations as the speed-up $S$. We expect a minimum speed-up of $>1.5$ to justify the Tesla V100 GPU price to performance. We recorded the computational time to reach convergence for the ISSM CG CPU and PT GPU solver implementations (Fig. 4) for up to $2e^7$ DoFs tested. Across glacier configurations, we reported a speed-up of $>1.5$ on the Tesla V100 GPU. We reported a speed-up of approximately $7\times$ at $\sim 1e^6$ DoFs for the Jakobshavn glacier model. This high speed-up at $\sim 1e^6$ DoFs indicates PT GPU implementation's suitability to develop high spatial resolution ice-sheet flow models. We reported an exception for the Jakobshavn glacier model at $2e^7$ DoFs, speed-up of $0.28\times$. We suggest that readers compare the speed-up results reported in this study with other approaches to parallelization.

The PT method applied to solve nonlinear momentum balance equations is a memory-bound algorithm as described in Sect. 3.3. On the Tesla V100s, with the increase in DoFs, the profiling results indicated an increased utilization of the device's available memory resources, up to 85%, further confirming the memory-bounded nature. To assess the performance of the memory-bound PT CUDA C implementation, we employ the effective memory throughput metric defined in Sect. 3.3. We report the effective memory throughput to DoFs for the PT CUDA C single GPU implementation (Fig. 5). We observed a significant drop in effective memory throughput on both GPU architectures at DoFs $> e^7$, which explains the drop in speed-up. We attribute the drop partly to the non-optimal global memory access patterns reported in the LITEX and L2 cache. We identified data non-localities in ice stiffness and strain rate computations involving element-to-vertex connectivity and vice versa as the primary source. For optimal or fully coalesced global memory access patterns, the threads in a warp must access the same relative address. We are investigating techniques to reduce the mesh non-localities and allow for coalesced global accesses.



The reported peak memory throughput for the GPU hardware NVIDIA Tesla V100 and NVIDIA Ampere A100 was 785 GB s$^{-1}$ and 1536 GB s$^{-1}$, respectively. The peak memory throughput only reports the memory transfer speed for performing memory copy operations. It represents the hardware performance limit. Across glacier model configurations for the DoFs chosen in this study, the PT CUDA C implementation achieves a maximum of 23 GB s$^{-1}$ for the Tesla V100 and 58 GB s$^{-1}$ for the Ampere A100. Thus the PT CUDA C implementation reaches 3% and 4% of peak hardware value on the Tesla V100

and the Ampere A100, demonstrating the potential for increase in effective memory throughput through better GPU resource exploitation. Future studies will include techniques to improve memory throughput and increase data locality.

Minimizing the memory footprint is critical to the memory-bounded algorithm's performance, better speed-ups, and increased ability to solve large-scale problems. Due to insufficient memory at $1e^8$ DoFs with Pine Island Glacier model configuration, we could neither execute the standard CPU implementation on a four 18-core Intel Xeon Gold 6140 processor with 3TB of RAM per vertex nor the PT GPU implementation on the Tesla V100 GPU architecture. However, we could execute the

PT GPU implementation on a single Ampere A100 SXM4 featuring 80GB onboard memory, further confirming the need to keep the memory footprint minimal for high-spatial-resolution models.

In this preliminary study, we tested up to an estimated $2e^7$ DoFs needed to maintain a spatial resolution of ∼1 km or better in grounding line regions for Antarctic and Greenland-wide ice flow models. Future studies would involve extending

the PT CUDA C implementation from (i) regional scale to ice-sheet scale and (ii) 2-D SSA to 3-D Blatter-Pattyn Higher-order (HO) approximation. To extend the PT CUDA C implementation to ice-sheet scale, we will choose the damping parameter $\gamma$, nonlinear viscosity relaxation scalar $\theta_\mu$, and transient *pseudo* time step $\Delta\tau$ carefully. The shared elliptical nature of the 2-D SSA and 3-D HO partial differential equations (Gilbarg and Trudinger, 1977; Tezaur et al., 2015) confirms the PT method's ability to solve the 3-D HO momentum balance applied to unstructured meshes. The overarching goal is to diminish spatial

resolution constraints at increased computing speed for improved predictions of grounding line migration at the ice-sheet scale.

## 5   Conclusions

Recent studies have implemented techniques that keep computational resources manageable at the ice-sheet scale while increasing the spatial resolution dynamically in areas where the grounding lines migrate during prognostic simulations (Cornford et al., 2013; Goelzer et al., 2017). In terms of computer memory and execution time, the computational cost associated with

solving the momentum balance equations to predict the ice velocity and pressure is a primary bottleneck (Jouvet et al., 2022). This preliminary study introduces a PT solver, applied to unstructured meshes, that leverages the GPU computing power to alleviate this bottleneck, as mentioned above. Coupling the ice velocity and pressure simulations executed on the GPUs with ice thickness and temperature simulations executed on the CPUs can provide an enhanced balance between speed and predictive performance.

The objective of this study was to investigate the PT CUDA C implementation for unstructured meshes and application to 2-D SSA approximation. For both Jakobshavn Isbræ and Pine Island glacier models, the number of nonlinear PT iterations to converge for a given number of vertices, $N$, scales in the order of $\approx \mathcal{O}(N^{1.2})$ or better. We observed an exception at $3e^7$





degrees of freedom (DoFs) for the Pine Island glacier model; optimal solver parameters are unidentifiable. We assessed and compared the performance of PT CUDA C implementation with a standard CPU implementation using two metrics: price and
power consumption to performance. The single Tesla V100 GPU is 1.5 times the price of the two Intel Xeon Gold 6140 CPU processors. The power consumption of the PT CUDA C implementation was approximately one-seventh of the standard CPU implementation for the test cases chosen in this study. We expect a minimum speed-up of $>1.5$ to justify the Tesla V100 GPU price to performance. We report the performance (or the speed-up) across glacier configurations for DoFs tested to be $>1.5$ on a Tesla V100. We identified an exception at $2e^7$ DoFs for the Jakobshavn Isbræ model, speed-up of $\sim0.28$. We attribute this
drop in the speed-up to non-optimal global memory access patterns reported in the LITEX and L2 cache. We are investigating techniques to reduce the mesh non-localities and allow optimal global memory accesses.

The study is a first step toward leveraging GPU processing power for accurate polar ice discharge predictions. The insights gained from this study will benefit efforts to diminish spatial resolution constraints at increased computing speed. The increased computing speed will allow running ensembles of ice-sheet flow simulations at the continental scale and at high resolution,
previously not possible, enabling quantification of model sensitivity to changes in future climate forcings. These findings will significantly benefit process-oriented and sea-level-projection studies over the coming decades.

*Code availability.* The current version of FastIceFlo is available for download from GitHub at: https://github.com/AnjaliSandip/FastIceFlo (last access: 1 March 2023) under the MIT license. The exact version of the model used to produce the results used in this paper is archived on Zenodo (https://doi.org/10.5281/zenodo.7689935), as are input data and scripts to run the model and produce the plots for all the simulations
presented in this paper. The PT CUDA C implementation run on a CUDA-capable GPU device.

*Author contributions.* **AS** developed PT CUDA C implementation, conducted the performance assessment tests described in the manuscript followed by data analysis, manuscript edition. **LR** provided guidance on the early stages of the mathematical reformulation of 2D SSA model to incorporate the PT method, supported the PT CUDA C implementation, manuscript edition. **MM** reformulated 2D SSA model to incorporate the PT method, developed the weak formulation, wrote the first versions of the code in MATLAB and then C. All authors
participated in the writing of the manuscript.

*Competing interests.* Ludovic Räss is on the Geoscientific Model Development editorial board.

*Acknowledgements.* We acknowledge University of North Dakota Computational Research Center for computing resources on Talon cluster and are grateful to David Apostal and Aaron Bergstrom for technical support. We thank NVIDIA Applied Research Accelerator Program for the hardware support. We thank NVIDIA solution architects Oded Green, Zoe Ryan and Jonathan Dursi for their thoughtful inputs. LR
thanks Ivan Utkin for helpful discussions and acknowledges the Laboratory of Hydraulics, Hydrology and Glaciology (VAW) at ETH Zurich



for computing access on the Superzack GPU server. CPU and GPU hardware architectures were made available through the first author's access to the University of North Dakota's Talon and NVIDIA's Curiosity clusters.



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
