# Peer review of "Graphics processing unit accelerated ice flow solver for unstructured meshes using the Shallow Shelf Approximation (FastIceFlo v1.0.1)"

_Geoscientific Model Development, 2023_

## Referee Comment (RC2)

**Review of "GPU accelerated ice flow solver for unstructured meshes using the Shallow Shelf Approximation (FastIceFlo v1.0)"**

Dan Martin

July 2023

**1 Overview**

This article presents algorithm and implementation details to port the SSA solver for the ISSM ice sheet model to GPUs, focusing on the nonlinear momentum solve which takes up the vast majority of computational effort in most ice sheet models. The authors do a good job of explaining their mathematical formulation and present a nice set of real-world test cases to demonstrate performance.

One thing the authors should make clearer is the set of differences (progress) from the previous work by Räss, et al and this work. I think I have a sense of it, but it would be best if it were clarified in the introduction. ("Previous work (Räss 2020, Räss 2019, etc) did xxxx. The work presented here builds upon that by ..." or something like that.)

I do have a set of fairly minor issues which I think should be addressed prior to publication (enumerated below), but once those have been addressed, I support publication in GMD.

**2 Specific Points**

1. line 30-31: "The performance of CPUs is slowly leveling off" – You could also mention the power-consumption issues pushing us toward GPUs here.

2. line 39: As I mention below, I don't think you should use the word "inertial" here

3. Eqn 3: Do you add any regularization to address the singularity when the strain rate is 0?

4. line 80: You're not really including the full inertial terms here, just a partial time derivative. The "inertial terms" are the nonlinear terms in the material derivative ($u \cdot \nabla u$). I think you're better off just saying that

you add a pseudo-time to allow iteration to a consistent steady state. (I realize this is what it was called in previous references; if I'd reviewed those, I'd have been equally pedantic for them :-) )

5. line 89: Since your pseudo time-step is spatially variable, do you have a sense of how uneven the convergence is?

6. line 98: I'd suggest "the stable explicit CFL time step", or "the explicit CFL-stable time step", etc. to reinforce that stability is driving this choice.

7. Eqn 10: I don't think this is correct – assuming that $H$ is spatially varying (i.e. a non-boring ice sheet), using the chain rule on the RHS of Eqn 9 results in an extra term: $4\mu \frac{\partial v_x}{\partial x} \frac{\partial H}{\partial x}$. Maybe it can be ignored, but I think there needs to be some justification of that.

8. Eqn 11: I'm confused by this, since it seems like you're including an extra $\Delta \tau_W A_x^{old}$ term in the update that would accumulate as it evolves. In a true oscillating wave that would likely get canceled out as it evolves, but this seems like it won't tend toward oscillatory wave solutions. Can you discuss that?

9. line 135 – I think it would be helpful to reference back to Eqn 4 here.

10. Figure 3: Is it possible to include (likely in an additional figure) some sort of plot of norm(residual) (i.e. du/dt) vs. iteration to illustrate how this method performs? Is it a linear convergence, or something better? (in the end, you're comparing against a more-standard iterative method where one would plot residual vs. iteration).

11. line 192: What do you mean by "arithmetic precision"? 32 vs 64? (what number are you using for $n_p$?)

12. line 200: Can you include a table with the optimal parameters here?

13. line 202: Can you describe what you mean by "optimal solver parameters are unidentifiable"? Are you completely unable to solve the problem? or is it simply that you can't identify optimal parameters (in which case you could still have a result)?

14. line 203: "in the following steps" – Perhaps this phrase is a relic of an edit?

15. line 205: (side note) I've also found that the presence of dynamically important ice shelves (like that in the PIG case) can drastically affect performance in the solution of the momentum balance.

16. line 213: Don't we really care about the integrated power needed to solve this system? (i.e. Watt-hours or Watt-seconds, etc? vs the power, which is a rate...)

17. line 253: "confirms" – I don't think it confirms so much as suggests...

18. line 255: Maybe say "ice sheet evolution" instead of "grounding-line migration"?

19. line 280: I'd probably say "not practical" or "not possible without extreme computational resources" vs. "not possible" (after all, we now have exascale computers (well, one exascale computer, at least)).

20. References: The links to the DOIs are messed up here, with repetitions of "https://doi.org"

**3 Typos and grammar fixes**

1. line 50: "West" should be capitalized in West Antarctica.

2. line 50: There is an extra space after "read"

3. line 71: "require to"

4. line 82: "allowing us to..."

5. line 103: "process to diverge" → "process diverging"

6. line 113: "allows us to aggressively..."

7. line 114: "method scale to" → "method scale as"

8. line 123: "limiter" → "limit"

9. Eqns 14, 17, 18 are badly formatted

10. line 165: "West Antarctica"

11. line 175: "As a CPU" → "For the CPU comparison" or something like that

12. line 177: missing space after "enabled"

13. line 187: "of PT CUDA: → "of the PT CUDA"

14. line 194: should be a colon after 8, not a semicolon, I think.

15. line 209: "the NVIDIA"

16. line 210:"The Power"

17. line 211: Should it be "sheet: thermal..."? (colon vs. semicolon)

18. line 266: "the 2-D SSA..."

---

## Author Response (AR1)

**Authors' response**

September 2023

**1 In response to Anonymous Reviewer 1's comments**

- Line 37: The authors state,"...the traditional way of solving the governing equations of ice-sheet flow, such as the finite-element analysis, is not adapted to GPUs as they cannot handle large sparse matrices and linear solvers." GPUs can handle large sparse linear solvers. There are vender specific libraries such as cuSolverSP. For multi-GPU iterative solvers see PETSc, Trilinos and Hypre.
  *We have revised the statement accordingly.*

- Lines 213-215: The authors state, "...the power consumption of the PT GPU implementation was approximately one-seventh of the traditional CPU implementation..." This comparison is misleading. A V100 GPU also requires a CPU which consumes power. One would need to include the power of the CPU for a fair comparison. A variation of this statement is also in the abstract (lines 13-14) and conclusion (line 271).
  *We have omitted power consumption as a performance metric. In the conclusion, we include a brief discussion stating our preliminary findings on the power consumption comparison.*

- Line 230: Where are L1TEX and L2 cache reported from?
  *We reported the L1TEX and L2 cache from NVIDIA's Nsight Compute generated reports for each ice flow simulation. The reports have been uploaded in the associated GitHub repository –*
  *https://github.com/AnjaliSandip/FastIceFlo/tree/master/output/Nsight Compute Reports*

- Lines 235-236: Where are the peak memory throughputs reported from? Was this an additional study performed?
  *We posted the code to determine the peak memory throughputs in the associated GitHub repository –*
  *https://github.com/AnjaliSandip/FastIceFlo/blob/master/scripts/memcopy.cu*

- Lines 238-240: 3-4% of measured peak memory throughput is very low and would indicate a latency bound kernel. These numbers should be

verified with a profiler such as nvprof or nsight.

*We have included the NVIDIA Nsight Compute Reports, which lists the measured memory throughput for the glacier model configurations run on the NVIDIA Tesla V100 GPU in the associated GitHub repository. The corresponding reports on the A100 GPUs are not, as we can no longer access the NVIDIA curiosity cluster. The differences between the measured, effective, and peak memory throughputs have been clarified in the revised manuscript.*

**1.1 Technical comments:**

- Equation 2: epsilons should be defined
  *In equation 2, the $\dot{\varepsilon}_{SSA}$ on LHS has been defined to be 2D SSA strain rate. The $\epsilon$ terms on the RHS have been replaced with partial derivatives of ice velocity with respect to x and y.*

- Equation 14: not on the same line
  *It is now equation 16 and is on the same line*

- Equation 14: $\epsilon_w$ should be defined. Should this be $\Delta$ w?
  *$\epsilon_w$ has been replaced with $\nabla \mathbf{w}$,*
  *the expression now reads $\int_\Omega 2H\mu\dot{\varepsilon}_{SSA} \cdot \nabla \mathbf{w} \, d\Omega$*

- Figure 2: It would be good to also show the meshes used in the study to better understand the mesh quality.
  *The meshes used in this study have now been uploaded in the associated GitHub repository –*
  *https://github.com/AnjaliSandip/FastIceFlo/tree/master/docs/Element Size*

- Line 187: "performance of PT" should be "performance of the PT"
  *The phrase "performance of PT" has been replaced with "performance of the PT".*

- Line 200: The values for the damping parameter, nonlinear viscosity relaxation scalar, and transient pseudo time step are not provided for the study.
  *We listed the range within which the nonlinear viscosity relaxation scalar and the damping parameter were varied to maintain linear scaling and solution stability on the lines following equations 8 and 14, respectively. We listed the formula to determine the pseudo transient time step in equation 7. The table with the optimal values has been included in the associated GitHub repository –*
  *https://github.com/AnjaliSandip/FastIceFlo/blob/master/README.md*

- Line 207: References are required for the price values of the Tesla V100 GPU and Xeon Gold 6140 CPU used in this study. A comparison between the prices is also written in the abstract (lines 12-13) and conclusion (line

270). References should be added there as well.
*The reference for the Intel Xeon Gold 6140 CPU price has been included. The price of one NVIDIA Tesla V100 SXM2 card at the time of purchase, as quoted by the vendor (dfennell@piergroup.com), was 7500 dollars.*

- Line 209: It would be good to have a reference for the NVIDIA System Management Interface used in this study and a version number.
  *We have included the NVIDIA System Management Interface driver version number in the revised manuscript. Additional details have been provided in the associated GitHub repository –
  https://github.com/AnjaliSandip/FastIceFlo/tree/master/output/TeslaV100*

- Line 211: A reference is also required for the hardware specification sheet used to acquire the thermal design power of the Intel Xeon Gold 6140 processor.
  *We have now included a reference for the hardware specification sheet.*

- Lines 219-222: A table or graph of speedups would help highlight the values obtained in the study.
  *We have now included a table in the revised manuscript listing the speedups, Table 1.*

**2 In response to Dan Martin's comments**

One thing the authors should make clearer is the set of differences (progress) from the previous work by Räss, et al and this work. I think I have a sense of it, but it would be best if it were clarified in the introduction. ("Previous work (Räss 2020, Räss 2019, etc) did xxxx. The work presented here builds upon that by ..." or something like that.)
*We have now included a statement in the introduction that implements this suggestion.*

**2.1 Specific Points:**

- line 30-31: "The performance of CPUs is slowly leveling off." You could also mention the power-consumption issues pushing us toward GPUs here.
  *Thank you, we have implemented the suggestion.*

- line 39: As I mention below, I don't think you should use the word "inertial" here
  *We have revised the statement accordingly.*

- Eqn 3: Do you add any regularization to address the singularity when the strain rate is 0?
  *Yes, we regularize our viscosity formulation in the GPU implementation by capping it at $1e^5$ to address the singularity when the strain rate tends*

*toward zero.*

*$etan[ix] = min(exp(rele * log(eta_{it}) + (1.0 - rele) * logf(etan[ix])), eta_0 *$*
*$1e^5)$*

*We include a statement indicating the same in the revised manuscript.*

- line 80: You're not really including the full inertial terms here, just a partial time derivative. The "inertial terms" are the nonlinear terms in the material derivative ($u \cdot \nabla u$). I think you're better off just saying that you add a pseudo-time to allow iteration to a consistent, steady state. (I realize this is what it was called in previous references; if I'd reviewed those, I'd have been equally pedantic for them :-) )
  *We have revised the statement accordingly.*

- line 89: Since your pseudo time-step is spatially variable, do you have a sense of how uneven the convergence is?
  *The spatially variable pseudo-time step speeds up the convergence by allowing for slow-moving regions to reach a steady state in a limited number of time steps compared to what would be required if we were using the smallest time step over the entire domain.*

- line 98: I'd suggest "the stable explicit CFL time step" or "the explicit CFL-stable time step," etc., to reinforce that stability is driving this choice.
  *The phrase has been modified from "explicit CFL time step" to "explicit CFL-stable time step" here and in Figure 1.*

- Eqn 10: I don't think this is correct – assuming that H is spatially varying (i.e. a non-boring ice sheet), using the chain rule on the RHS of Eqn 9 results in an extra term: $4\mu \frac{\partial v_x}{\partial x} \frac{\partial H}{\partial x}$ Maybe it can be ignored, but I think there needs to be some justification of that.
  *Thank you for reporting this. Yes, it is incorrect, as H is spatially variable. We updated the equations to correct the problem.*

- Eqn 11: I'm confused by this, since it seems like you're including an extra $\Delta\tau_W A_x^{old}$ term in the update that would accumulate as it evolves. In a true oscillating wave that would likely get canceled out as it evolves, but this seems like it won't tend toward oscillatory wave solutions. Can you discuss that?
  *Thank you for flagging this. We indeed had introduced the wrong time step definition, and there was confusion between two different ways of highlighting the analogy of our implementation with the work of Frankel 1950 (second-order Richardson method). We now clarified this in the paper and also updated the equations.*

- line 135: I think it would be helpful to reference back to Eqn 4 here.
  *We have included a reference to Equation 4.*

- Figure 3: Is it possible to include (likely in an additional figure) some sort of plot of norm(residual) (i.e. du/dt) vs. iteration to illustrate how this

method performs? Is it a linear convergence, or something better? (in the end, you're comparing against a more standard iterative method where one would plot residual vs. iteration).

*Yes, we have now included a plot of error residual vs. nonlinear pseudo-transient iterations for each glacier model configuration (Figure 4). For both Jakobshavn and Pine Island glacier models, the number of nonlinear pseudo-transient iterations required to converge for a given number of vertices N scales in the order of $\approx O(N^{1.2})$ or better.*

- line 192: What do you mean by "arithmetic precision"? 32 vs. 64? (What number are you using for $n_p$?)
  *$n_p$ represents the size of the data type (double - 8 bytes)*

- line 200: Can you include a table with the optimal parameters here?
  The table with the optimal parameters has been included in the associated GitHub repository –
  https://github.com/AnjaliSandip/FastIceFlo/blob/master/README.md

- line 202: Can you describe what you mean by "optimal solver parameters are unidentifiable"? Are you completely unable to solve the problem? Or is it simply that you can't identify optimal parameters (in which case you could still have a result)?
  *We were unable to identify solver parameters that result in convergence (or meet the chosen stopping criterion) at $\approx 3e^7$ degrees of freedom (DoFs) for the Pine Island glacier model. We will investigate this in the next steps.*

- line 203: "in the following steps" – Perhaps this phrase is a relic of an edit?
  *We have omitted the phrase "in the following steps", now on line 209.*

- line 205: (side note) I've also found that the presence of dynamically important ice shelves (like that in the PIG case) can drastically affect performance in the solution of the momentum balance.
  *Thank you for sharing this.*

- line 213: Don't we really care about the integrated power needed to solve this system? (i.e. Watt-hours or Watt-seconds, etc? vs the power, which is a rate...)
  *We have omitted power consumption as a performance metric. In the conclusion, we include a brief discussion stating our preliminary findings on the power consumption comparison.*

- line 253: "confirms" – I don't think it confirms so much as suggests...
  *We have replaced "confirms" with "suggests", now on line 252.*

- line 255: Maybe say "ice sheet evolution" instead of "grounding-line migration"?
  *We have replaced the phrase "grounding line migration at the ice-sheet scale" with "ice sheet evolution", now on line 253.*

- line 280: I'd probably say "not practical" or "not possible without extreme computational resources" vs. "not possible" (after all, we now have exascale computers (well, one exascale computer, at least)).
  *We have rephrased the sentence accordingly, now on line 280.*

- References: The links to the DOIs are messed up here, with repetitions of "https://doi.org"
  *The links to the DOIs have now been fixed.*

**2.2  Typos and grammar fixes:**

- line 50: "West" should be capitalized in West Antarctica.
  *"West" is now capitalized in West Antarctica*

- line 50: There is an extra space after "read"
  *We have omitted the extra space, now line 64.*

- line 71: "require to"
  *We have omitted the sentence containing this phrase in the revised manuscript.*

- line 82: "allowing us to..."
  *We have omitted the sentence containing this phrase in the revised manuscript.*

- line 103: "process to diverge" → "process diverging"
  *We have replaced the phrase "processed to diverge" with "process diverging".*

- line 113: "allows us to aggressively..."
  *We have replaced the phrase "allows to aggressively" with "allows us to aggressively".*

- line 114: "method scale to" → "method scale as"
  *We have replaced the phrase "method scale to" with "method scale as".*

- line 123: "limiter" → "limit"
  *We have replaced the word "limiter" with "limit".*

- Eqns 14, 17, 18 are badly formatted
  *We have resolved the formatting errors in equations 14, 17, and 18, now labeled 16, 19, and 20.*

- line 165: "West Antarctica"
  *We have replaced the phrase "west Antarctica" with "West Antarctica".*

- line 175: "As a CPU" → "For the CPU comparison" or something like that
  *We have rephrased the sentence accordingly, now on line 181.*

- line 177: missing space after "enabled"
  *We have added the space after the word "enabled", now on line 183.*

- line 187: "of PT CUDA: → "of the PT CUDA"
  *All occurrences of the phrase "of PT CUDA" have been replaced with "of the PT CUDA".*

- line 194: should be a colon after 8, not a semicolon, I think.
  *We have replaced the semicolon with a colon, now on line 199.*

- line 209: "the NVIDIA"
  *All references to NVIDIA have been revised to "the NVIDIA".*

- line 210: "The Power"
  *In the revised manuscript, this line is now in the conclusion section. We have revised the line to reflect the change.*

- line 211: Should it be "sheet: thermal..."? (colon vs. semicolon)
  *In the revised manuscript, this line is now in the conclusion section. We have revised the line to reflect the change.*

- line 266: "the 2-D SSA..."
  *All references to 2-D SSA have been revised to "the 2-D SSA".*

---

## Author Response (AR2)

**Authors' response**

**November 2023**

- The paper can be accepted for publication subject to the inclusion of the table with parameter values as an appendix rather than hidden in the github repository. This was asked by the two initial reviewers and reiterated during the re-review.

  *The table with the optimal parameter values has now been included in the appendix. A reference to the table has been included on line 205 of the revised manuscript.*